# Genomic insights from a final Bronze Age community buried in a collective tumulus in an Urnfield settlement in Northeastern Iberia
Marina Bretos Ezcurra [1,2], Adam B. Rohrlach [2,3], Luka Papac[2], José Ignacio Royo Guillén[4], Rodrigo Barquera [2], Fabiola Gómez Lecumberri[5], Rafael Laborda Lorente[1,6], Roberto Risch[7], Johannes Krause [2], Jesús V. Picazo Millán [1] ✉, Wolfgang Haak [2] ✉ & Vanessa Villalba-Mouco [2,8] ✉

The transition from the Bronze Age (BA) to the Iron Age (IA) on the Northeastern Iberian Peninsula is characterized by the emergence of cremation as the main funerary practice. Cultural attributes of a group, known as the Urnfield Culture, expanded from Central Europe to Northeastern Iberia during the Final Bronze Age (FBA), from ~1300 to ~850 cal BCE. Various hypotheses on the group's emergence exist, but cremations hinder DNA preservation. Here, we present genome-wide data from 24 inhumed individuals from a collective burial mound at the site of Los Castellets II (Spain), where inhumations and cremations co-occurred during the FBA, and one Early Iron Age (EIA) individual from Los Piojos (Spain). The results show that two source populations are required to explain the ancestry at Los Castellets II: one enriched in steppe-related ancestry and distantly related to Central European BA populations, and a second source similar to local Southeastern Iberian BA. Additionally, two-thirds of the individuals from the same collective tumulus were closely biologically related from 1st to >6th degree, with a man having the highest number of genetic relatives. We detected signs of inbreeding within the family group, all together suggesting the tumulus was used as a family mausoleum.

European prehistoric genetic diversity has been shaped by key demographic events, with the last major transformation being the influx of ancestry from Pontic-Caspian steppe-related pastoralists from 3000 BCE to 2000 BCE, e.g.,[1–3]. However, in the Bronze Age (~2200–850 cal BCE), diverse cultural groups emerged and disseminated throughout the continent, which also generated debates about cultural diffusion versus cross-regional migrations that have not yet been explored in detail by archaeogenetic methods. Specifically, the final phase of the Bronze Age in Europe (~1300–850 cal BCE) is characterised by the appearance of the Urnfield culture, which extends westwards, from Central Europe to the Northeast of the Iberian Peninsula, and which established cremation as a novel and widespread funerary ritual

among different shared traits[4–6] (Fig. 1A). Cremations continued as the main funerary treatment in the ensuing Iron Age (IA) societies, where only infant individuals[7–10] and some adults[11] did not receive this treatment.

Throughout this text, we use the term Bronze Age (BA) to refer to the entire chronological period (2200–850 BCE) in Iberia. We divide this broad horizon into two phases: the Early-Middle BA (EMBA), which we will refer to as the date range between ~2200 and ~1400 BCE, and the Late Bronze Age (LBA), the period between ~1400 and ~850 BCE. The final LBA, which coincides with the appearance of cremations, will be referred to as Final Bronze Age (FBA), covering a chronological period between ~1300 and ~850 BCE for the region under study (Fig. 1C). In this way, we aim to resolve

[1]Institute for Research in Environmental Sciences of Aragon (IUCA), University of Zaragoza, Calle de Pedro Cerbuna 12, Zaragoza, Spain. [2]Department of Archaeogenetics, Max Planck Institute for Evolutionary Anthropology, Deutscher Platz 6, Leipzig, Germany. [3]School of Biological Sciences, University of Adelaide, Adelaide, SA, Australia. [4]Dirección General de Cultura y Patrimonio, Gobierno de Aragón, Avenida de Ranillas 5 D., Zaragoza, Spain. [5]Independent Researcher, Zaragoza, Spain. [6]Paleoymás, Pol Empresarium C/retama 17 24C, La Cartuja Baja-Zaragoza, Spain. [7]Department of Prehistory, Universitat Autònoma de Barcelona, Barcelona, Spain. [8]Institute of Evolutionary Biology, CSIC-Universitat Pompeu Fabra, Barcelona, Spain. ✉e-mail: jpicazo@unizar.es; wolfgang_haak@eva.mpg.de; vanessa_villalba@eva.mpg.de

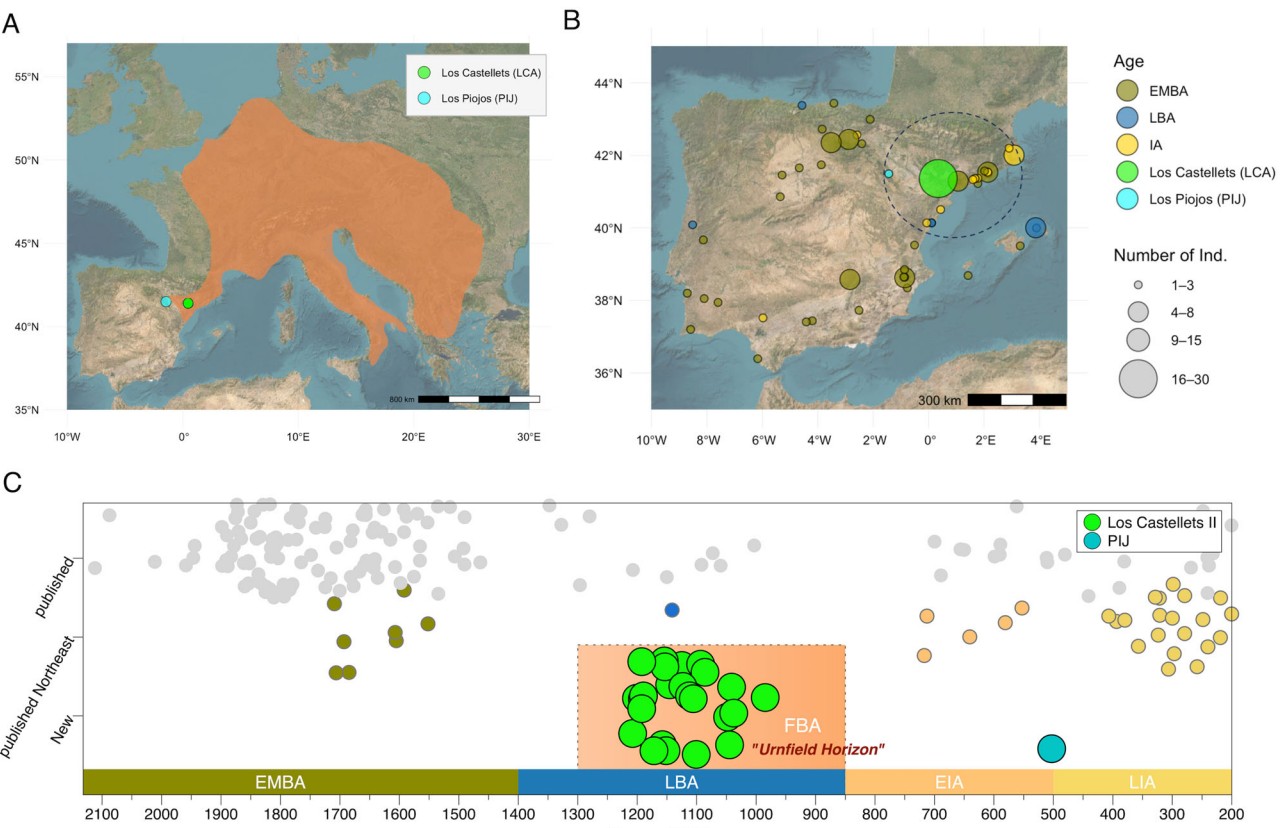

**Fig. 1 | Geographic locations and timeline of sites under study. A** Geographic distribution of cremation burial practices associated with the Urnfield complex horizon c. 1300 BCE adapted from ref. 74, with coloured circles denoting the new sites Los Castellets II (LCA) and Los Piojos (PIJ). **B** Map of Iberia with coloured circles denoting sites from where genome-wide data is available and the new sites Los Castellets II (LCA) and Los Piojos (PIJ), with the Iberian Northeast indicated by the dashed circle, and point size indicating samples sizes for each site. **C** Chronological time scale for published individuals from the Iberian Peninsula (dark green for EMBA and blue for LBA individuals from the Northeast of Iberia) and the new individuals analysed in this study (green and blue circles) (Supplementary Data 2.1). Directly radiocarbon-dated individuals are plotted according to their mean calibrated date (2-sigma range), and a jitter option within their specific time range was applied for individuals that were dated by archaeological context. Random jitter was only applied to the Y axis.

the discrepancies that exist in different geographic areas within and outside of the Iberian Peninsula concerning the horizon defined as the Middle BA.

The cremation and subsequent deposition of human remains inside urns present significant challenges for bioarchaeological research. From a molecular perspective, cremation drastically reduces the chances to recover collagen and/or ancient DNA[12], which are the main sources of data for direct chronological, dietary, and demographic inference, adding further difficulties in the study of this period. However, recent advances in the analysis of cremated remains, including morphological, osteometric, and histological approaches, have substantially increased the information obtained from these materials[13]. In addition, improved wet lab protocols for Strontium ratio ($^{87}Sr/^{86}Sr$) analysis allowed researchers to infer individual mobility patterns during life in contexts where both inhumation and cremation practices coexisted[14,15]. There are only a few Bronze Age sites known where cremations and inhumations potentially co-occurred, e.g., ref. 16, and the site of Los Castellets II presented here is one such example (Supplementary Information 1, Supplementary Fig. 1).

**The 'Urnfield culture' in the Northeast of the Iberian Peninsula**

The emergence of the cremation practices associated with the 'Urnfield culture' in Northeastern Iberia has been discussed through different archaeological perspectives. The first cremation necropolises were studied by Bosch-Gimpera, who, although after initially suggesting a possible local southeast/Argaric influence, changed his explanation for the Urnfield culture in the northeast of Iberia as a migration from the Hallstatt culture, located in Central Europe, as the primary source[17,18]. The Hallstatt

hypothesis was adopted by several researchers during the 20th century who discussed different scenarios ranging from large-scale migrations/invasions[19], to short-distance population movements from across the Pyrenees around ~1300 cal BCE, but emphasising the active role of local BA groups[20]. Currently, researchers consider autochthonous development and population continuity as the prevailing model that facilitated the transition from the FBA to the IA at the local scale. This model rules out substantial population movements (folk migration) but does not exclude limited contacts with other groups[7,21]. Indeed, the entire concept of a unique 'Urnfield culture' horizon in Iberia has been questioned. Some of its defining characteristics have been shown to have emerged and spread at different times: for example, channel-decorated pottery dates to around 1300 cal BCE, while cremation cemeteries did not appear until around 1100–1000 cal BCE[22,23].

From a genomic perspective, little is known about the FBA groups from the Iberian Peninsula. So far, relatively few individuals have been published (n = 7, excluding those with low coverage) and none of these individuals were from a region influenced by the Urnfield group[3,24,25] (Fig. 1B, C). A general increase in steppe-related ancestry from BA to IA has been detected in northern Spain and today's France[3,10,25]. One possible explanation for this phenomenon in Iberia was that the appearance of the Urnfield group could have been associated with excess steppe-related ancestry, consistent with a higher proportion of the latter observed in Central Europe[3]. However, so far, no direct genetic link has been identified between central European LBA populations, which leaves the question of the origin of this increase in steppe-related ancestry unresolved.

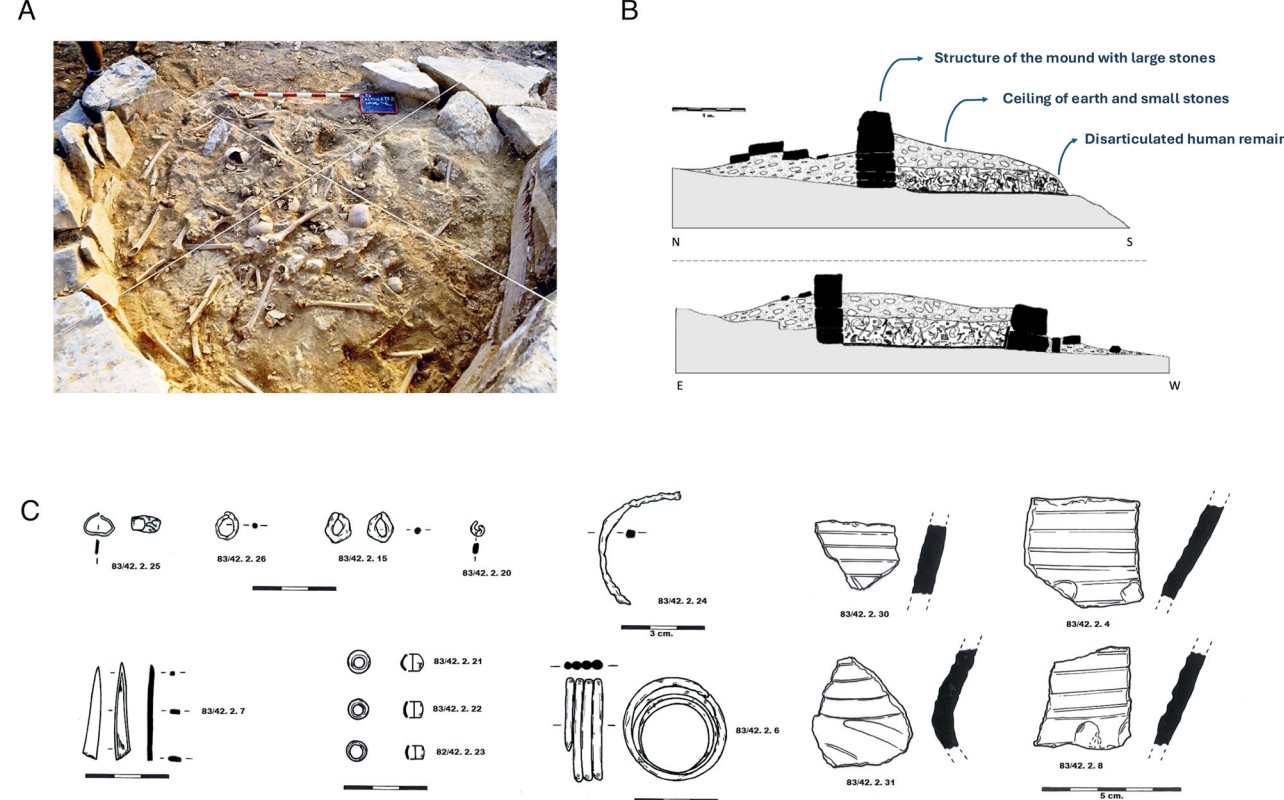

**Fig. 2 | Archaeological features and context of Tumulus 2, Los Castellets II.**
**A** Photograph of Tumulus 2 at the moment of excavation showing the distribution of skeletal human remains extracted from [75]. **B** Archaeological profile of the tumulus, from North (N) to South (S), and East (E) to West (W). **C** Archaeological drawing of the grave goods recovered from inside the collective funerary tumulus: small bronze beads, bronze bracelet, bone awl fragment, and channel-decorated ceramic fragments (author: Royo Guillén).

Here, we analyse individuals from the unique site of Los Castellets II, where the two burial practices of inhumation and cremation show a chronological and a cultural *continuum* and might have coexisted for some time, shedding light onto the Urnfield communities from Northeastern Iberia. For comparison, we have also produced data from an Early IA (EIA) individual from the northeastern Iberian site of Cueva de los Piojos, directly dated to 760–431 cal BCE (Supplementary Information 1, Supplementary Data 1.1).

### Los Castellets II: a unique LBA necropolis in Northeastern Iberia
The archaeological complex of Los Castellets is located in the municipality of Mequinenza, near the confluence of the Segre and Ebro rivers in the northeast of today's Spain (Fig. 1B). The site extends over two rocky spurs, separated by a deep ravine. A settlement with material culture dating from the EMBA (~1850–1550 BCE) to the Iberian culture (Late Iron Age, LIA; ~500–100 BCE) is located on the hill situated farthest east, including characteristic items from the local Urnfield culture. An extensive necropolis of cremation mounds dating to the FBA/IA was discovered next to the settlement, named Los Castellets I. The western spur is called Los Castellets II, and is largely occupied by a mixed necropolis of inhumation and cremation mounds that spans from the FBA (~1300 BCE) to the LIA (~500 BCE). Both inhumations and cremations show a chronological and cultural *continuum* with a minimal overlap in time ~900 and ~800 BCE (Supplementary Information 1, Supplementary Fig. 1), standing out as the only example on the Northeastern peninsula where this ritual diversity is observed in a narrow time window. From ~800 BCE on, the practice of inhumation ended, and cremations became more common/predominant, lasting until the end of the Early Iron Age (~500 BCE), when the necropolis was abandoned. The settlement itself continued at least until the 4th-3rd century when Iberian groups (referred to as *Illerguetes* in Classical sources)

were present in this area. The first centuries of the settlement (from ~1300 to 700 BCE), along with many inhumation and cremations, were ascribed to the local Urnfield culture by Royo Guillén[26,27] (Supplementary Information 1).

There is a correlation between collective/individual burial and inhumation/cremation practices, since all cremations seem to be of single individuals, and the few examples of collective burials are inhumations. Collective burial mounds, understood as the successive, non-contemporaneous deposition of individuals over time in the same burial space, represent a very ancient funerary tradition that was practised in the region since the Neolithic[28]. Tumulus 2, the main funerary structure analysed here, consists of a collective inhumation burial, containing at least 30 individuals dating to ~1200–827 cal BCE (Supplementary Data 1.1) and had channel-decorated ceramics among the recovered grave goods (Fig. 2)[29]. This provides an opportunity to genetically characterise this group of individuals that shared the same funerary space, chronological time frame, and material culture with the first communities that cremated their dead. Most of the individuals from Tumulus 2 were not found in an anatomical position due to the displacement of body parts. Individuals more proximal to the entrance were less disturbed than the ones located further back (Supplementary Information 1) which was probably associated with generating more space inside the mound for later burials, often with small temporal differences. However, secondary inhumation of some individuals (the collection of human bones buried somewhere else and later redeposited into the funerary mound) was also suggested. We selected 23 adults from an estimated minimum number of individuals (MNI) of 30 (76%)[29]. The total number of samples was obtained from the selection of skull fragments with petrous bones and/or mandibles with teeth. One additional individual included in the study (LCA031) was buried in tumulus 27, another collective burial with at least three individuals[30].

## Objectives

The partial chronological overlap of inhumations and cremations at the FBA necropolis of Los Castellets II and the fact that both types of funerary treatments share certain material culture traits, makes this site unique among those studied to date in northeastern Iberia (Supplementary Information 1, Supplementary Fig. 1). The main objectives of this study were i) to determine, whether we can infer changes in genetic ancestry during the FBA, as seen in the material contexts and funerary practices, ii) to shed light on the different hypotheses regarding the spread of the so-called Urnfield group in northeastern Iberia, and iii) to describe the potential genetic sources that contributed genetic ancestry to FBA groups in comparison with those from the preceding EMBA from Iberia. We also investigated the biological relatedness, signals of inbreeding, and the potential social practices and burial organisation of the FBA inhumation Tumulus 2, by integrating genetic and archaeological data. However, it will not be possible to answer other questions – such as whether these individuals with a traditional collective funerary ritual are representative of the whole FBA society – until a larger sample of FBA and IA individuals is analysed.

To aid in our genetic ancestry analyses and modelling, we report and co-analyse a new Early IA (EIA) individual from the northeastern Iberian site of Cueva de los Piojos, directly dated to 760–431 cal BCE (Supplementary Information 1, Supplementary Data 1.1).

## Results

### The genetic dataset

We initially screened 38 ancient human samples (petrous bones and teeth) for DNA preservation. Ancient DNA (aDNA) was extracted and converted to single-stranded non-UDG libraries by the automated robotic pipeline established at the Archaeogenetics Departments (MPI-EVA Jena/Leipzig) (see Methods). aDNA preservation was evaluated by shallow shotgun screening of ~5 million single-end reads on a Hiseq4000 Illumina platform, followed by an assessment of % endogenous human DNA (minimum of 0.1%), average read length between 40 and 75 base pairs (bp), and characteristic aDNA damage patterns at the 5'-end (see Methods, Supplementary Data 1.1 and 1.2). Libraries fulfilling these criteria were retained for in-solution hybridization capture of 1.24 million informative sites (1240k SNP panel) in the human genome[31], mitochondrial DNA capture[1] and Y-chromosome capture[32], separately (Supplementary Data 1.2, 1.3, 1.4 and 1.5). In total, we generated genome-wide data from 25 individuals from two different archaeological sites from the northeastern Iberian site of Los Castellets II (FBA site, $n = 24$) and Los Piojos (EIA site, $n = 1$) (Supplementary Information 1, Supplementary Data 1.1).

After capture, the endogenous human DNA content for the on-target 1240k SNPs yielded percentages ranging from 0.55 to 80.11% (Supplementary Data 1.2). We estimated genetic sex following[33] (Supplementary Data 1.6) and assessed contamination rates in both autosomal data using ANGSD (in males)[34] and AuthentiCT (for both genetic sexes)[35], and mitochondrial data using ContamMix (for both genetic sexes)[36]. The results indicated minimal contamination for the studied individuals, with estimates <3% for nuclear DNA in males and <3% for mitochondrial DNA in all individuals (Supplementary Data 1.1).

We used BREADR[37], KIN[38], and READ[39] to identify potential duplicate samples, as they come from collective burial contexts. We found that two samples, LCA005.A (left petrous bone) and LCA009.B (lower left third molar) showed a pairwise mismatch rate consistent with being sampled from the same individual (the most likely scenario) or being identical twins, and thus, subsequently merged the data, treating them as a single individual, separating LCA009.A and LCA009.B. LCA010.AB and LCA007.A were found to be related in the 1st-degree, and we excluded the lower coverage sample LCA007.A from the population genetic analyses (Supplementary Information 2, Supplementary Fig. 2). Apart from these two cases, we also detected several pairs at different degrees of relatedness higher than the 1st-degree (described below and in Supplementary Data 1.7 and 1.8).

Overall, our final dataset consisted of 25 individuals ranging from 3158 to 722,429 targeted 1240 k SNPs (see Methods, Supplementary Data 1.1).

We selected individuals with >40,000 SNPs from the 1240k panel when analysing them separately, leading to the exclusion of nine individuals with insufficient SNP coverage and retaining 16 for population genetic analyses. We merged our dataset with the published AADR V54.1.p1 data repository, which contains genotype data from ancient and modern human populations[40]. Supplementary Data 2.1 contains the individual IDs and the population labels used for the demographic inference. For the biological relatedness analyses, we included individual pairs with more than 1000 shared SNPs, which allowed us to test a combination of 240 pairs. Finally, we used ancIBD[41] on individuals with ~400,000 SNPs to calculate IBD sharing between pairs of individuals (see Methods and Supplementary Data 1.9).

### Demographic insights from LBA and Iron Age northeastern Iberia

In order to explore the genetic affinities between the newly genotyped FBA and EIA individuals from Northeastern Iberia with other geographically and chronologically close groups, we performed principal component analysis (PCA) calculated on a set of modern-day West Eurasian populations genotyped on the Human Origins (HO) SNP panel[42] onto which ancient individuals were projected (Fig. 3A). We observed that our FBA individuals from Los Castellets II fall into a position that partially overlaps with published Iberian EMBA and LBA individuals but outside of the variation of southeastern EMBA groups, such as those associated with El Argar (2200–1550 BCE) or the Ibero-Levantine/Valencian BA, which carry less "steppe-related ancestry" than EMBA groups from the North[42]. Furthermore, the new EIA individual from Cueva de los Piojos takes a higher PC2 value than the FBA individuals from the Los Castellets II site. We explored whether the shifts in PC space were based on differences in data treatment using different genotyping methods and were ultimately able to exclude this possibility (Supplementary Information 3.2, Supplementary Fig. 3).

Olalde and colleagues[3] showed that steppe-related ancestry had increased during the transition from the BA to the IA, and suggested that the spread of the Urnfield culture from Central Europe could have been a possible cause for this change. However, the increased level of steppe-related ancestry was evaluated using the entire IA dataset available at that time, none of which was related to the core area of the Iberian urnfields. Therefore, our FBA dataset from an Urnfield necropolis, with contemporaneous inhumations and cremations and similar material culture in both types of funerary treatments, allows us to re-evaluate these conclusions using several approaches. Firstly, we formally tested if we can detect an increase in the amount of steppe ancestry at a regional level in Northeast Iberia by quantifying the amount of this ancestry using the distal qpAdm model proposed by ref. 25 optimised to keep the standard errors as low as possible. This particular model uses WHG (Western Hunter gatherers), EEF (Eastern Early Farmers) and OldSteppe (Steppe pastoralists) sources, and a genetically close set of outgroups (Fig. 3B) (Supplementary Data 2.1). We ran this model individually and as a group for NE_Spain_EMBA, Los Castellets II, NE_Spain_FBA_Mortorum, NE_Spain_IA_Early (including the new EIA site Los Piojos) and NE_Spain_IA_Late to evaluate changes in the proportion of ancestral sources over time (Fig. 3B and Supplementary Data 2.2). Our results show an increase in the percentage of steppe-related ancestry between EMBA and FBA, although the increase detected is not significant (Wilcoxon rank-sum test; $p = 0.06935$). Using the PC2 coordinates as a proxy for steppe-related ancestry from the same individuals confirms a positive but non-significant trend between the two periods (Wilcoxon rank-sum test; $p = 0.099$). Significant differences in the proportion of steppe-related ancestry are only observed when comparing EMBA vs EIA and LIA from Northeastern Iberia (Supplementary Information 3.2, Supplementary Fig. 4, Supplementary Data 2.3). Furthermore, $f_4$-statistics of the form $f_4(Mbuti, test; Russia\_Samara\_EBA\_Yamnaya, Turkey\_N)$ are consistently positive indicating an increase in steppe-related ancestry over the FBA, but are not statistically significant, applying $|Z| \geq 3$ (Supplementary Data 2.4, Supplementary Information S3.3, Supplementary Fig. 5). These results, obtained specifically from the northeastern part of the Iberian Peninsula, suggest patterns similar to those obtained by Olalde[3] for the entire Iberian Peninsula. Altogether, we can not rule out the hypothesis that the increase in

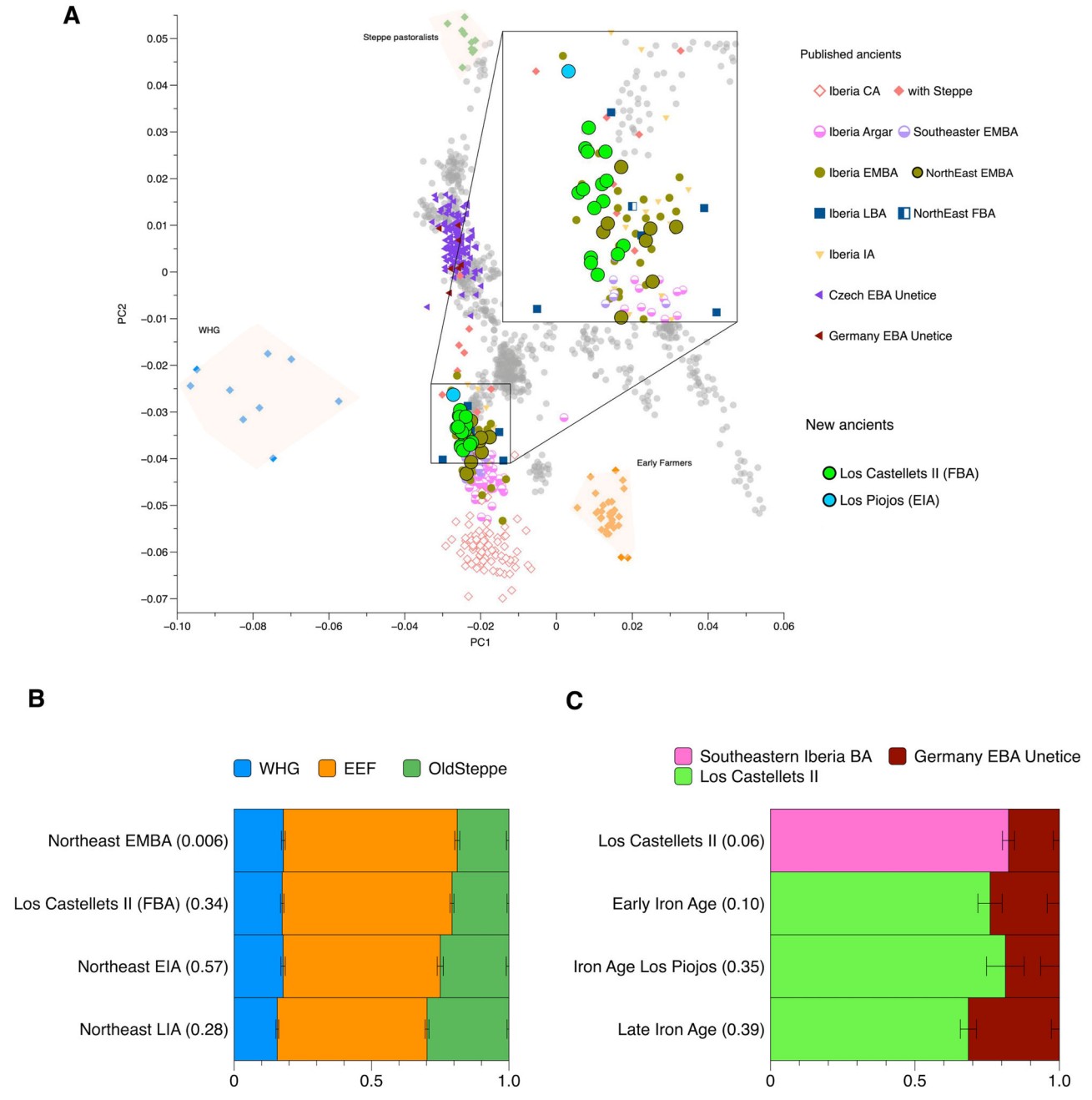

**Fig. 3 | Population genetic analyses of BA and IA populations from the Iberian Peninsula. A** PCA of present-day west Eurasian individuals (grey dots), with ancient individuals from Iberia and other regions of interest projected onto the first two principal components. **B** Northeastern EMBA, FBA, EIA, and LIA groups modelled as a distal three-way mixture model using the sources Early European Farmers (EEF), Western hunter-gatherers (WHG), and Old_Steppe (Supplementary Data 2.1), *p*-values given in brackets in the y-axis labels (Supplementary Data 2.2). **C** Northeastern LBA, EIA and IA populations modelled using qpAdm and proximal sources, *p*-values given in brackets in the y-axis labels (Supplementary Data 2.2). Non-supported models and their corresponding *p*-values are reported in Supplementary Data 2.5, 2.7. Error bars indicate one standard error.

steppe-related ancestry could have begun in the FBA. However, additional data from FBA archaeological sites across different geographic regions are needed to document a potentially earlier increase of steppe-related ancestry and whether it was associated with specific cultural groups.

We next aimed to explore more proximal models in order to test different hypotheses inferred from archaeological evidence. First, we tested the hypothesis of autochthonous developments in which the null hypothesis assumes population (i.e., genetic) continuity through time. Specifically, we tested whether Los Castellets II is genetically indistinguishable from the preceding population from the same geographical area (NE_Iberia_EMBA or N_Iberia_EMBA). However, using a standard set of outgroups

(Supplementary Data 2.5) Los Castellets II could not be successfully modelled with NE_Iberia_EMBA (*p*-value = 7.29E−05) nor N_Iberia_EMBA (*p*-value = 1.48E−05) as single source, which rejects the null hypothesis of population continuity in northeastern Iberia between EMBA and FBA (Supplementary Data 2.5). The single individual from the published Túmulo Mortorum with, presumably, a chronology slightly earlier than or similar (1300–1000 BCE) to Los Castellets II, could be modelled as 100% NE_Iberia_EMBA, but possibly due to a limited resolution and reduced statistical power when analysing a single individual (Supplementary Data 2.5).

We then explored alternative hypotheses that involved mixture models of two sources of genetic ancestry consistent with archaeological hypotheses

about additional genetic influences from the Urnfield period and other potential demographic scenarios. Firstly, we tried to model Los Castellets II as a mixture of the preceding NE_Iberia_EMBA and Central Europe BA or Central Mediterranean BA groups as second sources (the territorial limits of the expansion of the Urnfield culture), but did not obtain a satisfactory model fit ($p$-values > 0.05) (see all combinations in Supplementary Data 2.6).

Since our ancestry modelling rejected genetic continuity between NE_Iberia_EMBA and the newly reported FBA individuals from Los Castellets II, and because adding additional ancestry from Central Europe or Central Mediterranean groups did not improve the model fit, we decided to specifically test if the ancestry could be explained by gene flow from southeastern Iberian EMBA groups. Early scholars such as Gimpera suggested a southern Iberian influence in the FBA groups from the northeast[17]. We thus grouped individuals from the late archaeological phase of the Argaric sites La Almoloya and La Bastida and individuals belonging to the Ibero-Levantine/Valencian BA along the Mediterranean coast (La Horna, Cabezo Redondo, Peñón de la Zorra, and Puntal de los Carniceros) and merged these under the population label "Southeastern_Iberia_EMBA" (Supplementary Data 2.1). We then tested if this group could be used as a single source of ancestry for FBA individuals, or in combination with other BA sources in two-way mixture models. We modelled Los Castellets II as Southeastern Iberia EMBA plus a list of potential sources including local and non-local BA groups (Supplementary Data 2.7). We observed that adding Southeastern_Iberia_EMBA systematically improved the model fit. For example, we obtained a good model fit when using Southeastern_Iberia_EMBA and Central Europe BA as two sources of ancestry. However, the model with Southeastern_Iberia_EMBA and Local_EMBA (either NE_Iberia_EMBA or N_Iberia_EMBA) as two sources was not supported (Fig. 3C, Supplementary Data 2.7). We then performed rotating qpAdm models including both Southeastern_Iberia_EMBA and NE_Iberia_EMBA as either outgroups or sources, and the results support Southeastern_Iberia_EMBA as the best-fitting source population (Supplementary Information S3.4).

In sum, these results suggest that the genetic ancestry represented by Southeastern_Iberia_EMBA individuals is currently the best proxy for the Iberian ancestry found at Los Castellets II. This model also fits for the published N_Spain_FBA individuals from El Espinoso (Asturias), although the ancestry proportion represented by Southeastern_Iberia_EMBA is lower than in Los Castellets II, and this group is represented by only two individuals (Supplementary Data 2.7 and S2.8). It has been shown that Southeastern_Iberia_EMBA is the group with the least steppe-related ancestry in Iberia, and additional gene flow from central/eastern Mediterranean populations was suggested[42]. Our results suggest that direct gene flow from the Central Mediterranean is not necessary and that local Southeastern Iberia EMBA is a fitting source to model in the FBA population represented by Los Castellets II.

Overall, our proximal models indicate that the genetic change between the tested EMBA groups and FBA groups could be driven by the influx of Southeastern Iberia EMBA ancestry to Northern Iberian regions. This result is consistent with the Y-chromosomal lineage R1b-P312 found in 100% of the male individuals from Los Castellets II (all individuals are consistent with R1b-Z198 with varying degrees of resolution due to sample quality), which is also most common in the preceding Iberian BA (Supplementary Data 1.1).

However, Southeastern_Iberia_EMBA ancestry does not fully explain the genetic makeup of the individuals from Los Castellets II. Additionally, a small proportion of ancestry best represented by central European BA groups with higher levels of steppe-related ancestry is required as a second source. Since we do not find any characteristic Y-chromosomal haplogroups from Central Europe, the Central European BA ancestry must have come through closer groups (e.g., southern LBA France), which in turn would have received gene flow from Central European BA groups. Patterns of female exogamy detected during BA in Europe could also explain subtle genetic shifts without introducing new Y-chromosomal haplogroups[43–45].

Finally, we were not able to model the EIA individual from Los Piojos (PIJ001) as 100% ancestry from Los Castellets II FBA individuals ($p$-value = 0.03) (Supplementary Data 2.5). Using our FBA individuals plus additional ancestry from Central European BA individuals instead yielded a better model fit, which suggests continuous gene flow from Central Europe from FBA to EIA in Northeastern Iberia (Fig. 3C, Supplementary Data 2.6).

## Insights into the social patterns of a LBA Iberian community

The kinship practices of Iberian FBA communities have not yet been explored due to a lack of sufficient data from this time period. Only few data are available from different sites (El Espinoso, Asturias, $n = 3$; El Sotillo, Álava, $n = 1$; Túmulo de Mortorum, Castellón, $n = 2$), and no intra-site study has been reported so far.

Here, we were able to co-analyse 24 adult individuals from the FBA tumuli site of Los Castellets II (23 from tumulus 2 and one from tumulus 27). We estimated the degree of relatedness among the individuals buried in the same tumulus (see Fig. 4A, B, Supplementary information 1, Supplementary Data 1.7, Supplementary Fig. 2). Notably, we found a high number of relatives: one pair of 1st-degree relatives (father and son), seven pairs of 2nd-degree relatives, two pairs of 3rd-degree relatives, five pairs of 4th-degree relatives, and at least 12 pairs of ~5th/6th-degree relatives. This implies that at least two-thirds of the sample were closely biologically related. Of the remaining eight individuals, being one from the separated Tumulus 27, who do not show a close biological link to any other individual, four were low coverage (<16,000 SNPs), and consequently, biological relatedness higher than 2nd-degree cannot be ruled out. This suggests that Tumulus 2 may have been used as a mausoleum by an extended family, where biological ties prevailed over other types of relationships.

We observed a roughly equal distribution of genetically female ($n = 11$) and male ($n = 13$) individuals inside the tumulus. However, we found more close biological links among male than among female individuals. We found one pair of 1st-degree relatives and seven pairs of 2nd-degree relatives. The first pair was composed of two adult men, LCA007 and LCA010, and the results obtained by KIN indicated a father-son relationship. However, this result needs to be taken with caution due to the low coverage of sample LCA007 (only 4500 overlapping SNPs) (Supplementary Data 1.7) and because the mitochondrial haplotypes are identical within the limits of our resolution (H2a in both individuals with 263 G, 750 G, 15326 G as observed polymorphisms) which makes the scenario of being siblings still possible. Conversely, no first-degree relationship was found between pairs of adult females or between adult male and female individuals. Additionally, we also identified eight pairs of individuals who were most likely related in the 2nd-degree. There were four pairs of adult male individuals (LCA003-LCA010, LCA003-LCA029, LCA020-LCA032 and LCA012-LCA028), two pairs of adult female individuals (LCA004-LCA014 and LCA004-LCA011), and two pairs of a male and an adult female individual (LCA033-LCA030 and LCA003-LCA004) (Fig. 4A). Two 2nd-degree pairs (LCA003-LCA029 and LCA004-LCA014) showed the same mitochondrial haplogroups (U5b1 for the first pair and U5a1c1a for the second pair), which is consistent with a 2nd-degree relationship on the maternal side. The remaining four did not match, and thus were consistent with a paternally-inherited relationship. Two pairs remain inconclusive (LCA020-LCA032 and LCA012-LCA028) (Supplementary Data 1.4). Using only IBD segments, we identify two pairs as very likely grandparent/grandchild (LCA003-LCA010 and LCA004-LCA011) and two pairs as possible avuncular or half-sibling relationships (LCA033-LCA030 and LCA003-LCA004).

The fact that we find more pairs of close male relatives (Fig. 4A), is supported by the average pairwise mismatch rate ($pmr$) obtained from READ when comparing female $vs.$ female, male $vs.$ male, and female $vs.$ male. On average, male individuals are more closely related to other individuals at the site than female individuals (Supplementary Data 1.8), which could be indicative of patrilineality and perhaps virilocality. However, when performing the locality test[42], the results obtained are not significant due to the presence of some women with relatives at the site ($p$-value = 0.08)

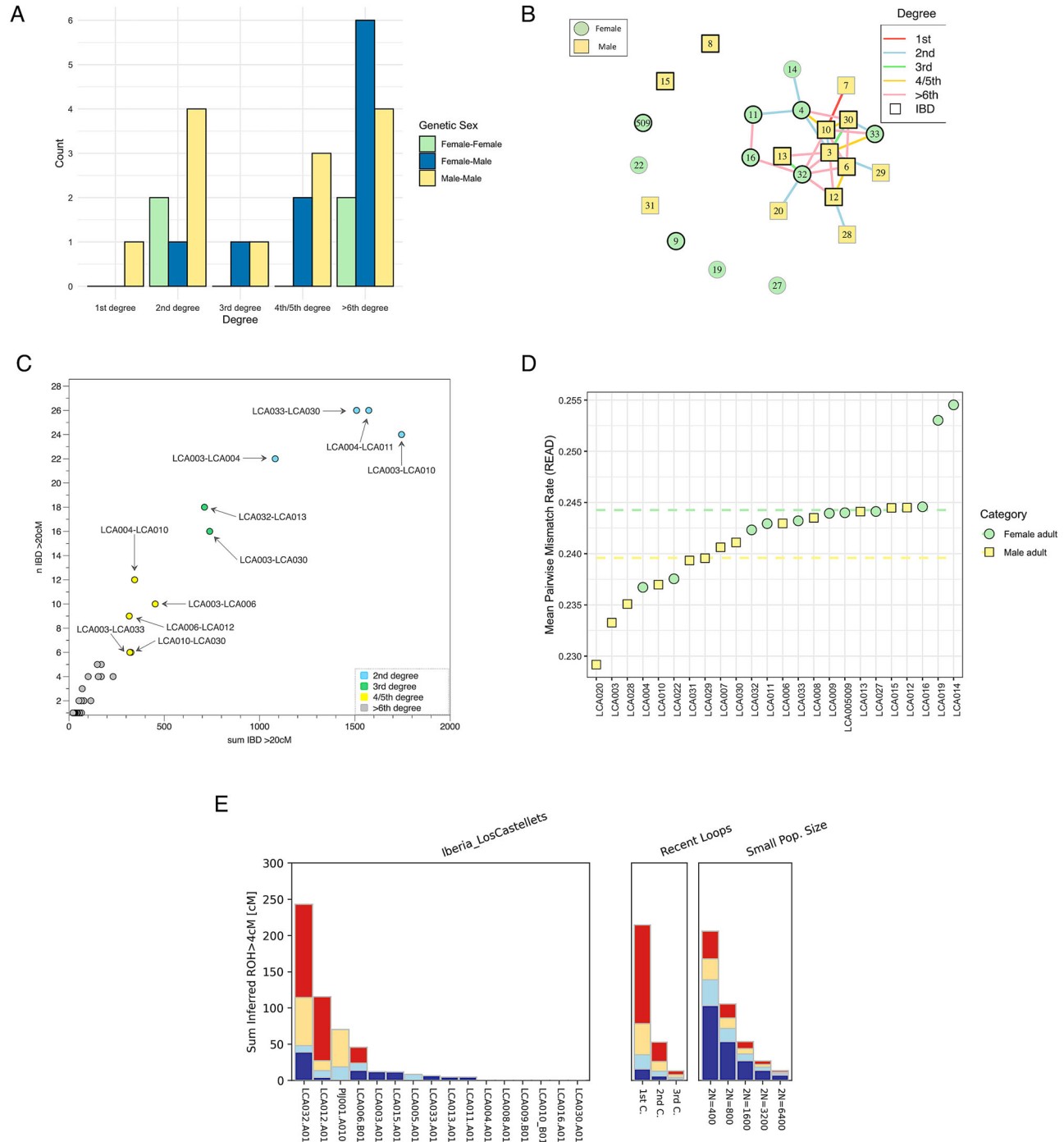

**Fig. 4 | Biological relatedness inference from Los Castellets II. A** Distribution of biological relatives within and between sexes in adult individuals and the network distribution of relatives estimated with BREADR and ancIBD (Supplementary Data 1.7, 1.9). **B** Network representing biological connections with the colours of lines connecting individuals indicating the degree of relatedness. Individuals with enough coverage for IBD are highlighted in black. **C** Identity-by-descent (IBD) results plotting the sum versus the number of shared chunks of IBD in window sizes of >20 cM. **D** Locality test: the mean pairwise mismatch rate (pmr) values of individuals compared to the pmr average of the tumulus 2 of Los Castellets II. Dashed lines indicate mean pmr in adult males (yellow) and adult females (green) (Supplementary Data 1.8). **E** Sum of all ROH segments measured with hapROH in individuals with >320,000 SNPs covered on 1240k panel (Supplementary Data 1.10).

(Fig. 4D). This scenario is similar to the one already observed during the Iberian EMBA, when clear indicators of patrilineal and virilocal practices were found at intra-site level but the statistical test remained non-significant[42]. A possible explanation for the lack of significance in this test is the inclusion of female individuals below the typical age for exogamy in the cohort of women. Additionally, at least eight individuals from the tumulus are related in the 2nd- to >5–6th-degree to the adult male individual LCA003, which places this individual in a central position of the network of

genetic relatives. Of the remaining seven individuals who are not closely related, two were male and five were female (Fig. 4B). Similar to the preceding EMBA period in Iberia, and generally in Europe, paternally-inherited lineages show a reduced variability of the Y chromosome and for Los Castellets II only R1b-PZ198 haplotypes (at different levels of resolution) were found (Supplementary Data 1.1). Although there are genetic indications suggesting virilocality in the case of Castellets II, the information from a single funerary tumulus is too limited to determine whether this is indeed

the case, or whether we are instead observing a less pronounced trend compared to earlier periods.

Finally, we observe elevated levels of parental background relatedness which are visible via long Runs of Homozygosity (ROH) (>20 cM) in two individuals, with at least one of these compatible with being the offspring of a first cousin union and one compatible with being the offspring of a second cousin union or higher (Fig. 4E) (Supplementary Data 1.10). Interestingly, both of these individuals are part of the biological family network. These results might indicate a particular endogamous practice within the network. However, we caution that due to the lack of extended data from other contemporaneous sites, testing for more generalised practice is not possible. A similar ROH accumulation pattern is observed in the EIA individual PIJ001 as well as in two EIA individuals from Las Eretas and Alto de la Cruz in Navarra[10], but was not observed during the preceding Chalcolithic and EMBA in Iberia[42].

## Discussion

Our population genomic results suggest the existence of a genetic discontinuity between the EMBA and our FBA of the individuals from the Northeastern Iberian site of Los Castellets II, which could be explained by a Southeastern Iberian influence and an additional contributing source of ancestry, which was enriched in steppe-related ancestry.

However, the limited number of genetic samples from geographically intermediate sites and chronologies do not allow us to define precisely where and when these potential connections were established. Although the majority of individuals who used to identify this southern affinity were associated with the El Argar and the Ibero-Levantine/Valencian BA cultures, these Southeastern Iberia BA individuals predate the newly reported FBA individuals by several centuries. During this time, population movements and genetic mixture could have been extensive and pervasive. From an archaeological point of view, the Southeastern Iberian ancestry observed in our genetic model is plausible in the light of the socioeconomic and political changes around 1550 BCE, one of the proposed reasons for the decay of El Argar, attested by archaeological research[46] which more drastically affected the organisation of the Iberian BA groups from Southern Iberia. Although our genetic data found stronger support from a local southwestern Mediterranean source rather than a central Mediterranean one, there is also archaeological evidence that suggests generally increased levels of interaction with the Mediterranean realm since 1300 BCE[47].

Additionally, ~17% of ancestry, as represented by central European BA proxies, is required to model the FBA individuals from Los Castellets II. However, our results support a more local/autochthonous phenomenon triggered by southeastern EMBA groups rather than exogenous/allochthonous contacts with central European BA sources. The central European BA ancestry is best understood as a genetic proxy, and this type of ancestry was likely contributed via a geographically more proximal source, that is largely unsampled LBA groups from southern France. The rationale behind this is that all of the Y-chromosomal lineages in Los Castellets II are consistent with the typical Iberian haplogroup (R1b-Z198), but none of these match the diversity found in Únětice groups from BA Central Europe[48].

From an archaeological perspective, the appearance of channelled pottery within collective burials reflects a connection of NE Iberia with Central Europe[49], whereas the funerary architecture resembles the megalithic traditions present in the same area since the 4th millennium BCE[50]. The construction of the tumulus at Los Castellets II led archaeologists to consider the persistence of "old" funerary practices for at least some groups, despite the adoption of new cultural elements during the FBA[26]. Thus, our integrative analysis of the archaeological contexts alongside genomic data rules out the ideas of an invasion or colonisation by people associated with the Urnfield phenomenon in the region during the FBA, at least for the inhumed individuals at the site of Los Castellets II. However, there is evidence of a continuous, yet subtle, cultural connection and genetic affinity to regions east of Iberia during the Bronze Age, possibly through intermediate groups.

A full appraisal of the population dynamics in this region during the FBA is still limited by the data available to date, and several aspects need to be taken with caution. Most importantly, the community that co-inhabited the site but practised cremation remains unsampled. Inhumations and cremations coexisted and shared material culture at Los Castellets but the cremation persisted longer while inhumations disappeared by ~800 BCE. The individuals from burial Tumulus 2 are only representative of this initial occupation phase of Los Castellets II, but we cannot rule out the possibility that contemporaneous and later cremated individuals may exhibit different genetic profiles. Furthermore, the settlement patterns in northeastern inland regions such as Los Castellets are markedly different from the coastal lands, the western plain, and mountainous areas[7,21], and subsequently the genetic history of FBA Iberia narrative could certainly be more complex.

The collective Tumulus 2 studied here stands out for its dimensions, the privileged location, and the number of individuals buried in it. Furthermore, the necropolis presents well-defined funeral sectors that have been associated with the existence of different family groups or clans[51]. In line with this hypothesis, Tumulus 2 was suggested to belong to a specific "family", understanding it as a related group but without specifying the type of relatedness (biological, cohabitation, or affinity, for example). Here, we reveal the biological relationships between the individuals buried in Tumulus 2, and the results are consistent with an extended biological family in which biological ties were emphasised alongside other forms of kinship. Male individuals had more biological links at the site than female individuals, which suggests a patrilineal form of social/funerary organisation, albeit not as pronounced as it was reported for other EMBA sites[43–45,52–55]. Comparable results have also been reported from Neolithic monumental and collective burials where the right to be buried in the monumental graves was primarily granted when a biological connection with a specific adult male existed[56,57], but also in flat necropolises[56,57]. The lack of 1st-degree relatives (only one pair of males) in the tumulus is unexpected and might reflect biases in sample preservation (e.g., skeletal remains of subadult individuals are less well preserved) or cultural practices with regard to the selection of individuals buried in these tumuli.

Additionally, we observe elevated levels of inbreeding in two individuals who belong to the extended biological group buried at Tumulus 2, and in the EIA individual from Los Piojos. Such levels of inbreeding have not been reported in previous chronologies for the Iberian Peninsula[42], but are observed in Early Iron Age Navarra[10], and thus could be a distinctive feature of extended biological families. The endogamous practices themselves have been suggested as a way to claim origins in hierarchical groups, with close-kin unions among elites in 'high-chiefdom' groups serving as a strategy for leaders to consolidate power and enhance their authority by deliberately challenging societal taboos[58–60]. This hypothesis finds support in the fact that the group was buried in a unique megalithic-like collective burial mound at the site, which might reflect a different social status and/or different beliefs. However, due to the limited data of FBA Iberian groups, we cannot exclude a more generalised practice during the LBA, and specifically the FBA.

Lastly, the particularities of this tumulus do not allow us to make inferences in other nearby tumuli from the same necropolises, as this is the only tumulus with such a large number of individuals, whereas others only contained a maximum of three individuals. The data obtained from the nearby Tumulus 27 (individual LCA031) were not of sufficient quality and quantity to estimate biological relatedness between all possible pairs, and no close relatives were found when testing was possible (>1000 overlapping sites) (Supplementary Information 1, Supplementary Data 1.7).

## Conclusions

The genetic analyses of the individuals buried in Tumulus 2 at Los Castellets II indicate that their ancestry derived from two distinct sources. One source exhibits a higher proportion of steppe-related ancestry compared to EMBA individuals from the same region, and can be explained by the contribution of Central European ancestry via other geographically proximal groups (e.g., LBA groups from Southern France). The other source is potentially linked to the southeastern Iberian EMBA, which yields a better fit than the preceding northeastern EMBA. With this new information and the archaeological information of the settlement, the appearance/presence of the Urnfield

culture in this region must be revisited. By providing a more realistic/nuanced framework for understanding the regional heterogeneity of FBA populations, these results are a valuable resource for future studies of populations with complex mixture and demographic histories, and and contribute to the longstanding debate regarding material culture versus human migration.

We were also able to provide new insights into burial and social patterns at the intra-site level. Our results show that the main funerary tumulus at Los Castellets II was used by an extended family, which was biologically more connected on the father's side than on the mother's side, where ~66% of the buried individuals were biologically related, and in which endogamy was practised to a certain extent. These results thus extend the knowledge about kinship practices during FBA in the Iberian Peninsula. Future research could integrate recent Strontium isotope protocols with genetic ancestry data to explore individual mobility in both cremated and non-cremated individuals, thereby offering a more comprehensive understanding of past population dynamics.

## Materials and Methods

### Materials
All individuals analysed in this study were obtained from the deposits of the Museum of Zaragoza with permission from the Cultural Heritage Department of Gobierno de Aragón. The archaeological sampling permit was issued on July 5th, 2019, by Ignacio Escuáin Borao, General Director of Cultural Heritage at the Gobierno de Aragón, and can be consulted in Supplementary Information 4.

The specimens have been returned to the storage facilities of the Museum of Zaragoza. The original archaeological codes are available in Supplementary Data 1.1, as well as with the archaeological materials, for consultation or replication analyses if needed.

### Radiocarbon dating
Four samples out of 33 from the same collective tumulus form Los Castellets II were radiocarbon dated, as well as the single individual analysed from Los Piojos cave. Radiocarbon dating was carried out from ultra-filtrated collagen and using accelerated mass spectrometry (AMS) at the Curt-Engelhorn-Zentrum Archäometrie gGmbH in Mannheim, Germany (raw and calibrated dates are available in Supplementary Data 1.1). All samples were calibrated using the IntCal20 calibration curve with Oxcal 4.4[61]. All [14]C dates in this study are consistent with the archaeological chronology based on stratigraphy and grave goods.

### Ancient DNA laboratory procedures
In total, we processed 17 petrous bones and 24 permanent molars retrieved from Tumulus 2 of the LBA site of Los Castellets II, in clean room facilities of the Max Planck Institute for Evolutionary Anthropology in Jena, Germany. Petrous bones were sampled with a minimally invasive method[62]. For teeth, the crown was separated from the root, and the inner pulp chamber was drilled out[63]. DNA was extracted and converted into non-UDG single-stranded libraries using an in-house robotic pipeline following a modified version of the Gansauge and Meyer[64] protocol.

All of the libraries were screened via shotgun sequencing of ~5 million reads on an Illumina HiSeq4000 sequencing platform using a single-end kit (SR 75). We then used nf-core/eager[65] for an initial quality assessment, based on the percentage of human mapped reads (Endogenous DNA%) and sequencing damage profiles[66]. Sixteen samples did not pass the threshold of >0.2% endogenous DNA. The remaining 28 samples with more than 0.2% of endogenous were selected for hybridization capture of 1.24 million informative sites ("1240k SNP capture")[31]. Enriched libraries were sequenced for 20 million reads on a HiSeq4000 Illumina platform using a single-read (SR 75) kit. In total, 24 individuals yielded sufficient genomic data for downstream population genetic analyses. Additionally, in-house mitochondrial capture ("MT capture"[1]) was performed for all individuals with less than 2,000 reads on the mitochondrial genome (Supplementary Data 1.3). For all male individuals, we also performed an in-house capture

assay for the Y-chromosome ("YMCA"[32],) which targets ~10.445 kB on the non-combining region of the Y chromosome (Supplementary Data 1.5).

### Sequencing data process
We used the nf-core/eager 2 pipeline v.2.3.5[65,67] to process captured sequencing reads. Under this pipeline, raw reads were trimmed for Illumina adapters using AdapterRemoval v2[68], mapped to the human reference genome hs37d5 using BWA v.0.7.12[69], and duplicates were removed using DeDup v.0.12.1[67]. aDNA damage at the 5' end was analysed using mapDamage v.2.0.9[66]. We called pseudohaploid genotypes on untrimmed BAM files using pileupCaller with the *singlestranded* mode (https://github.com/stschiff/sequenceTools). The numbers of SNPs covered at least once for the 1240k sites are reported in Supplementary Data 1.1.

### Ancient DNA authentication
All captured libraries yielded damage patterns characteristic of aDNA, which includes short DNA fragments and post-mortem deamination at the end of the molecules and showed on-target coverage ranging from 0.004X to 2.12X (Supplementary Data 1.2).

We determined genetic sex by calculating the coverage on the X and Y chromosomes and the autosomes using a bed file of the regions captured by the 1240k SNP array[33]. We did not observe signs of contamination of the mixing of both genetic sexes.

We estimate levels of contamination via the nuclear genome in males using ANGSD[34] and via the mitochondrial genome using ContamMix[36] (Supplementary Data 1.1). Additionally, we used AuthentCT to show that all captured libraries indicated negligible to low levels of contamination[35] (Supplementary Data 1.1).

### DNA reference datasets
We merged our final dataset with previously published datasets of ancient and modern individuals reported by Allen Ancient DNA Resource (AADR, V54.1)[40]. We used the "1240 K_HO" dataset for PCA analysis and the "1240K" dataset for the remaining genomic analyses. A list of individuals used and their population labels can be found in Supplementary Data 2.1.

### Genetic relatedness estimation
For the detection of close biological relatives, we used several methods, depending on the data quality, such as BREADR[37], KIN[38], and READ[39]. Samples LCA005.A and LCA009.B were identified as the same individual and therefore merged for downstream analysis. The results reported by these software packages can be found in Supplementary Data 1.7.

### Detection of Runs of Homozygosity (ROH)
We used hapROH[70] to detect ROH in individuals with >320,000 SNPs covered in the 1240k panel.

### Assignment of uniparentally-inherited haplogroups
Y-chromosome haplogroups for all male individuals were assigned using the manual assignment method of Y-haplogroup calling as described in ref. 32.

Trimmed reads from Shotgun, 1240k, and MT-capture were aligned to the revised Cambridge Reference Sequence (rCRS), and the consensus sequence was obtained using iVar (https://andersen-lab.github.io/ivar/html/manualpage.html). MT haplogroups were assigned using Haplogrep 3[71].

### Population genetic analysis
We performed principal component analysis (PCA) using the smartpca method included in the EIGENSOFT package (v6.0.1)[72] with the lsqproject and autoshrink options set to YES, and a reference dataset of contemporary populations from West Eurasia and North Africa from the Human Origins panel genotyped in the AADR[40]. We projected the newly genotyped ancient individuals onto the PCA, and plotted the first and second principal components (PC1 and PC2).

For Genetic admixture modelling analyses, we used qpWave and qpAdm from the ADMIXTOOLS v5.1 package (https://github.com/DReichLab), with the option "allsnps: YES". We used a standard set of outgroups (Ethiopia_4500BP.SG Spain_MLN France_MN Germany_BellBeaker Spain_HG Iran_GanjDareh_N England_N) which was extended following a rotating strategy by moving populations from the *sources* to the *outgroup* set when aiming to distinguish the better proximal source for our tested model.

## IBD sharing

IBD sharing analysis was performed using ancIBD[41] on individuals with more than 400 k SNPs and GP > 0.99 after imputation with GLIMPSE[73]. We used HapBLOCK43 to estimate IBD sharing. Imputed samples were merged, and the vcf_to_1240K_hdf command was employed to convert the VCF files into HDF5 format. The hapBLOCK_chroms command was then utilised to perform the IBD sharing analysis for each chromosome individually, using the default parameters.

## Statistics and reproducibility

The specifics of each statistical analysis and the software used are provided in the Methods and Supplementary Information. The number of individuals included in each analysis, along with the specific values, is shown in Supplementary Datas 1 and 2.

## Inclusion & Ethics

All necessary permits were obtained from the relevant institutions (see Supplementary Information 4), and analyses were conducted with respect for the archaeological context and in compliance with established ethical guidelines for the study of ancient DNA. The archaeological researchers, supported by the Cultural Heritage Department of the Gobierno de Aragón, approved and provided guidance on the sample selection and sampling protocols (see Supplementary Information 4).

## Reporting summary

Further information on research design is available in the Nature Portfolio Reporting Summary linked to this article.

## Data availability

Sequenced data in FASTQ and BAM format is accessible through the European Nucleotide Archive (ENA) under the project name PRJEB85840. Archaeological IDs, as well as the associated raw and calibrated radiocarbon dates, are available in Supplementary Data 1.1. All other data are available from the corresponding author upon reasonable request. The published ancient genomes co-analyzed in this study are available through the Allen Ancient DNA Resource (AADR), Version v54.1.p1. Numerical source data for graphs and charts can be found in Supplementary Data 1 and 2.

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

## Acknowledgements

We thank all members of the Archaeogenetics Department of the Max Planck Institute for Evolutionary Anthropology, especially the Population Genetics and PALEoRIDER groups, for scientific discussion about the data and population genomic analysis. We thank A. Wissgott and R. Radzeviciute for their support on the aDNA lab and sequencing tasks. We also thank K. Prüfer for the processing of the raw sequence data and L. Traverso, T. Lamnidis and H. Ringbauer of the Big Data group. We are indebted to Belén Gimeno for her support on sampling materials at the installations of Gobierno de Aragón, as well as all the institutions and archaeologists involved in the excavations, especially the University of Zaragoza and Museo de Zaragoza. This study was supported by the Max Planck Society and the European Research Council (ERC) under the European Union's Horizon 2020 Research and Innovation Programme Grant 771234-PALEoRIDER (to W.H.), and by the Spanish Ministry of Economy, Industry and Competitiveness project PID2022-140671NB-I00 (to J.V.P.). M.B. has a predoctoral scholarship funded by the Gobierno de Aragón and did a research stay at the MPI-EVA funded by the Fundación Ibercaja-CAI (2023). M.B. is a member of the regional government of Aragón Primeros Pobladores y Patrimonio Arqueológico del Valle del Ebro (P3A). M.B., J.V.P. and V.V.-M. are members of the Spanish project PID2022-140671NB-I00 (led by J.V.P.). V.V.-M. is supported by the grant "Ayudas para contratos Ramón y Cajal" (RYC2022-035700-I) funded by Ministerio de Ciencia, Innovación y Universidades. RR is supported by the ICREA Academia programme of the Generalitat de Catalunya. This article is part of the PhD thesis of M.B., supervised by J.V.P. and V.V.-M.

## Author contributions

This study was designed by W.H. and V.V.-M. Sample preparations and laboratory work were conducted by R.B., J.I.R.G., F.G.L., R.L. and V.V-M., and subsequent analyses were performed and discussed by M.B., V.V.-M., A.B.R., L.P., J.K. and W.H. The integration of the archaeological data was performed by M.B., J.V.P.M. and R.R. The original manuscript was prepared by M.B., V.V.-M., W.H. and A.B.R. and reviewed by all co-authors.

## Funding

## Competing interests

The authors declare no competing interests.
