## [Transparent Peer Review file · Communications Biology]

Genomic Insights from a Final Bronze Age Community buried in a Collective Tumulus in an Urnfield Settlement in Northeastern Iberia

Corresponding Author: Dr Vanessa Villalba-Mouco

Version 0:

Reviewer comments:

Reviewer #1

(Remarks to the Author)

In the manuscript "Genomic Insights from a Final Bronze Age Community buried in a Collective Tumulus in an Urnfield Settlement in Northeastern Iberia", Bretos Ezcurra et al. generate genome-wide data from 24 individuals from the collective burial of Los Castelletts II from the Iberian Late Bronze Age period. The transition from the Bronze Age to the Iron Age on the Northeastern Iberian Peninsula is characterized by the use of cremation as the main funerary practice, which complicates the recovery of ancient DNA. The interest of Los Castelletts II resides in the fact that inhumations and cremations co-occurred, allowing for exploring the Late Bronze Age period from a paleogenomic perspective and testing for changes in genetic ancestry during the Final Bronze Age period.

I think that the research question around which this manuscript is centered is interesting and the results are compelling. The manuscript is well written and the conclusions are well supported by extensive data analyses. It seems that the manuscript has benefited from previous reviews and the authors have already addressed the concerns of previous reviewers. This manuscript will be of interest to diverse fields, including paleogenomics, evolutionary biology and population genetics, and it is a good fit for Communications Biology. However, I think that there are some changes that could benefit the manuscript.

The "Urnfield culture" section should be rewritten to improve clarity. For example, some sentences are too long, which makes it difficult to follow the main idea.

Regarding the mtDNA results, it would be easier to understand the limitations of determining if two individuals have or do not have the same haplotype if mtDNA coverages are discussed, especially considering that individual LCA014 has a coverage of 0.2 X.

Please include the reference for the permit from the Cultural Heritage Department of Gobierno de Aragón.

A minor point, "from distant region" in line 522 should be "from a distant region".

Reviewer #2

(Remarks to the Author)

This manuscript presents genetic analyses of Late Bronze Age (LBA) and Early Iron Age (EIA) populations from northeastern Iberia, focusing on the unique burial site of Los Castelletts II, characterized by overlapping cremation and inhumation burial practices. The authors aim to address longstanding archaeological debates about cultural diffusion versus population migration, specifically related to the spread of the Urnfield culture. Through genomic analyses, they identify two ancestral components contributing to the site's genetic makeup—one local southeastern Iberian component and another carrying higher steppe-related ancestry potentially linked to Central European populations. The authors further investigate biological relatedness, kinship practices, and evidence of inbreeding within this population.

Major Comments

Strengths

- Historical and archaeological context: The introduction is thorough, clearly contextualizing the archaeological debates and significance of the Urnfield culture, providing a solid foundation for the analyses presented.
- The Castellets II site is unique and a good choice for exploring transitional burial practices, given the coexistence of cremation and inhumation (a rare thing to capture this transition of cultural practices)

Weaknesses

1. Limited statistical significance:

Many central claims rely heavily on statistically non-significant results. Statements such as:

"Our results show a positive increase in the percentage of steppe-related ancestry between EMBA and FBA, although the increase detected is not significant (Wilcoxon rank-sum test; $p = 0.06935$). Using the PC2 coordinates as a proxy for steppe-related ancestry from the same individuals confirms a positive but non-significant trend between the two periods (Wilcoxon rank-sum test; $p = 0.099$)."

Although the authors are careful to mention the insignificance of the statistical tests, they still seem to be drawing conclusions from these tests anyway. This undermines my confidence in the conclusions and I think this manuscript requires a more cautious interpretation and re-writing of many of the conclusions.

The authors extensively interpret subtle differences observed in PCA plots. I am not convinced by the trend observed in PCA. The qpAdm models despite acknowledging their limited statistical support.

2. Ambiguity and exploratory models:

Multiple qpAdm exploratory models are presented with limited clarity or justification regarding their selection and interpretation. The manuscript acknowledges:

"Firstly, we tried to model Los Castellets II as a mixture... but did not obtain a satisfactory model fit (p -values > 0.05)."

Clearer rationale and more explicit discussion on exploratory modeling would enhance the manuscript significantly. I was generally confused by the paragraphs describing all the different models used, I re-read it multiple times but struggled to make sense of why they tested certain models. I think more effort can be made to explain the rationale for the tests used, and maybe some of these results can be moved to a supplementary materials?

3. The discussion contains overly broad statements, like "Our population genomic results document..." These statements would benefit from mentioning which specific analyses they are referring to. Also highlighting limitations and uncertainty more transparently. Especially with all the different models mentioned in the results section, it was confusing what analyses the authors are referring to.

4. Sample size limitations: The authors themselves highlight limitations regarding sample size: "A full appraisal of the population dynamics in this region during the FBA is still limited by the data available to date, and several aspects need to be taken with caution." I commend the authors for being up front about this, but it also clearly limits the interpretation of their results. Unfortunately this remains to be a weakness in the manuscript as it stands, and is likely a main reason for the lacking statistical significance.

Minor Comments

- Figure 1 Issues:

- The legend and map symbols in Figure 1A are inconsistent, making it challenging to interpret accurately. Specifically referring to the line thickness of the stars.

- Figure 1B's star size is ambiguous. Clarifying whether star diameter or area corresponds to sample size would improve readability. Or perhaps not using stars for the size? The circles are much easier to interpret.

- Figure 1C shows discrepancies between timeline definitions in the figure and those in the text. Please make these consistent it was a confusing to me.

- Clarification of the number of samples included. (very minor point, possible typo)

- Initial statements suggest 25 individuals were analyzed, later adjusted to 24 due to merging duplicated samples:

"two samples... showed a pairwise mismatch rate consistent with being sampled from the same individual... thus subsequently merged... treating them as a single individual." But then later (L260) the authors mention 25 individuals used in the final dataset.

- Kinship Analysis:

- The kinship analyses largely confirm expectations (e.g., extended family burial), limiting their novelty somewhat:

"This suggests that Tumulus 2 may have been used like a mausoleum by a single extended family..." Am I missing something? I would imagine that a family mausoleum would have family members in it?

- Results indicating patrilineality lack statistical significance:

"...results obtained are not significant due to the presence of some women with relatives at the site (p -value = 0.08)." Again, I realize the manuscript is limited by sample sizes, but this result follows a similar trend of underwhelming statistical results.

(Remarks to the Author)

The paper provides genomic insights into a Late Bronze Age community in north-eastern Iberia, contributing to our understanding of population dynamics, mortuary ritual and mobility patterns of the period. The study will be conducted with scientific rigour and will be an important addition to the bioarchaeological literature.

While the paper presents valuable genetic data, some claims require further support:

Clarification on cremation analysis (line 46, abstract): The authors should specify on which samples aDNA analyses were performed, given the mixed nature of the context (both cremations and inhumations).

Correct the statement regarding the novelty of the cremation studies (lines 76-77): This statement should be revised to acknowledge the significant advances in cremation studies, including morphological, osteometric and histological investigations. References such as Gigante et al. (2021) should be considered.

Clarification of isotopic vs. genomic analyses (line 78): The current presentation of the state of the art is somewhat confusing. A distinction should be made between isotopic and genomic analyses. In addition, the authors should mention that radiogenic Sr isotopes are widely used for studies of individual geographic mobility. Recent publications (e.g. Esposito et al. 2023, Gigante et al. 2025) have successfully applied Sr ratio analysis in contexts where both inhumation and cremation practices coexisted.

Reorganisation of the chronological discussion (line 85): The section introducing chronological abbreviations should be moved earlier in the text to maintain narrative coherence, as the preceding discussion focuses on cremation and bioarchaeological studies.

Clarification of terminology (line 98): The term "historiographical perspective" does not seem appropriate. "Archaeological perspective" would be a more appropriate alternative.

Distinction between multiple and collective burials (line 173): Throughout the text, the authors use "multiple" and "collective" interchangeably when referring to the presence of several individuals in the same burial space. From a taphonomic perspective, however, these terms have different meanings: "multiple" implies synchronous burials, while "collective" suggests non-contemporaneous burials. The authors should revise the text to clarify the depositional sequence of the individuals studied.

The study has the potential to contribute to ongoing discussions of Final Bronze Age demographic patterns and social organisation in Iberia. However, its impact could be enhanced by a clearer articulation of its unique contributions, particularly with regard to cremation analyses, mobility studies, and the limitations of the dataset. The omission of mobility analysis for cremated individuals should be explicitly mentioned, and potential avenues for future research should be suggested.

Version 1:

Reviewer comments:

Reviewer #1

(Remarks to the Author)

The authors have taken into consideration all suggestions. I think the manuscript is suitable for publication in *Communications Biology*.

Reviewer #2

(Remarks to the Author)

Thank you for the effort you invested in revising the manuscript. The redesigned figures are clearer, and the slightly re-ordered Result section reads more smoothly. Nonetheless, I still feel that some central conclusions reach a bit beyond what the data securely demonstrate. Below I flag the points that, in my view, most need tightening to bring the narrative fully in line with the evidence.

1. PCA-based inference of Steppe ancestry

The manuscript still states: "Altogether, the PCA suggests an increased level of steppe-related ancestry in Iberian FBA and EIA individuals when compared with EMBA individuals from the same region."

Given the large overlap between EMBA and FBA clouds, I am not yet convinced that the modest shift in PCA space is biologically meaningful. Please either re-phrase to something more cautious (e.g. "is consistent with"), or omit the sentence and let the formal qpAdm / f-statistics carry the claim.

2. Use of non-significant trends

The revised text now reports Wilcoxon p-values of ≈ 0.07 – 0.10 for both Steppe ancestry estimates and PC2 scores, yet still frames these results as evidence of a "gradual increase."

At conventional thresholds these trends are not statistically significant. Please make this explicit and consider focusing

discussion on comparisons that do reach significance.

We agree that Olalde (2019) documents a statistically supported uptick in Steppe-rich ancestry between Bronze- and Iron-Age Iberia and tentatively links it to Urnfield movements. However, those data do not resolve whether the rise began earlier—within the Final Bronze Age—or specifically within the northeastern Urnfield heartland. Your dataset, not Olalde's, must provide that evidence, ideally with statistics stronger than a $p \approx 0.07$ Wilcoxon.

The manuscript is clearly improved, but its two main narrative pillars—(i) a PCA-based Steppe signal within the FBA and (ii) trends hovering just above the significance threshold—still need to be addressed in my opinion.

Thank you again for your revisions. I look forward to reviewing the next iteration.

Reviewer #3

(Remarks to the Author)

The authors have adequately addressed the issues raised in the minor revisions, including the incorporation of the requested bibliographic references. I find that the manuscript is now suitable for publication, and I have no further comments.

Version 2:

Reviewer comments:

Reviewer #2

(Remarks to the Author)

The authors have addressed the remaining issues, I have no further comments.

We would like to thank the reviewers and the editor for their thoughtful and constructive comments. We have carefully addressed each of the points raised and have revised the manuscript accordingly. Below, we provide a detailed, point-by-point response. Reviewer comments are presented in regular black, followed by our responses in blue italics. All changes made to the manuscript are highlighted in the revised version. The removed sections are shown with strikethrough formatting.

Reviewer #1 (Remarks to the Author):

In the manuscript "Genomic Insights from a Final Bronze Age Community buried in a Collective Tumulus in an Urnfield Settlement in Northeastern Iberia", Bretos Ezcurra et al. generate genome-wide data from 24 individuals from the collective burial of Los Castelletts II from the Iberian Late Bronze Age period. The transition from the Bronze Age to the Iron Age on the Northeastern Iberian Peninsula is characterized by the use of cremation as the main funerary practice, which complicates the recovery of ancient DNA. The interest of Los Castelletts II resides in the fact that inhumations and cremations co-occurred, allowing for exploring the Late Bronze Age period from a paleogenomic perspective and testing for changes in genetic ancestry during the Final Bronze Age period.

I think that the research question around which this manuscript is centered is interesting and the results are compelling. The manuscript is well written and the conclusions are well supported by extensive data analyses. It seems that the manuscript has benefited from previous reviews and the authors have already addressed the concerns of previous reviewers. This manuscript will be of interest to diverse fields, including paleogenomics, evolutionary biology and population genetics, and it is a good fit for Communications Biology. However, I think that there are some changes that could benefit the manuscript.

Many thanks, we appreciate the positive feedback.

The "Urnfield culture" section should be rewritten to improve clarity. For example, some sentences are too long, which makes it difficult to follow the main idea.

Thanks. We have rewritten some parts to make the reading clear. In lines 117-122, we split the sentence and added punctuation marks to improve the flow of the text. We have proceeded in the same way in lines 122-127, where a long sentence has been split into two separate ones. We believe that, following these adjustments, the entire paragraph is expressed clearly:

"This model rules out substantial population movements (folk migration) but does not exclude limited contacts with other groups (López Cachero, 2008a; Ruiz Zapatero, 1983). Indeed, the entire concept of a unique 'Urnfield culture' horizon in Iberia has been questioned. Some of its defining characteristics have been shown to have emerged and spread at different times: for example, channel-decorated pottery dates to around 1,300 cal BCE, while cremation cemeteries did not appear until around 1,100-1,000 cal BCE (López Cachero, 2011; Rodanés & Picazo, 2018)."

Regarding the mtDNA results, it would be easier to understand the limitations of determining if two individuals have or do not have the same haplotype if mtDNA coverages are discussed, especially considering that individual LCA014 has a coverage of 0.2 X.

Thanks. We noticed that we did not have a proper explanation of which samples were used to determine the mitochondrial haplogroup, that is either from the MT reads recovered from 1240k capture or via MT capture. We have now added a column in Table S1.4 with the MT depth coverage of each individual. We report MT haplogroups of individuals with more than 2500 MT reads, which corresponds with coverage ranging from 11.2 X to 107.9 X. MT-coverage. The depth coverage of MT reads from the 1240k capture by-catch is 39.5 X for LCA010 and 42 X for LCA007, and the haplogroup determination is H2a. Hence, there is no problem with coverage here but we mention that the haplogroups of two 1st-degree related males are the same based on matching haplotype-defining polymorphisms found in both individuals (which would be more likely in the scenario of siblings, although the results from KIN points to a father-son relation). This is still possible when not all polymorphisms are found or when the MT haplotype is very common, as in this case.

Please include the reference for the permit from the Cultural Heritage Department of Gobierno de Aragón.

Following your request we are in communication with the Editor to find a way to upload the document as a supplementary file. The permit does not have a specific number, but it is officially signed with the stamp of Cultural Heritage Department.

A minor point, "from distant region" in line 522 should be "from a distant region".

Thank you! The typo has been corrected and this paragraph has been moved from line 578 to 583.

Reviewer #2 (Remarks to the Author):

This manuscript presents genetic analyses of Late Bronze Age (LBA) and Early Iron Age (EIA) populations from northeastern Iberia, focusing on the unique burial site of Los Castelletts II, characterized by overlapping cremation and inhumation burial practices. The authors aim to address longstanding archaeological debates about cultural diffusion versus population migration, specifically related to the spread of the Urnfield culture. Through genomic analyses, they identify two ancestral components contributing to the site's genetic makeup—one local southeastern Iberian component and another carrying higher steppe-related ancestry potentially linked to Central European populations. The authors further investigate biological relatedness, kinship practices, and evidence of inbreeding within this population.

Major Comments

Strengths

- Historical and archaeological context: The introduction is thorough, clearly contextualizing the archaeological debates and significance of the Urnfield culture, providing a solid foundation for the analyses presented.
- The Castelletts II site is unique and a good choice for exploring transitional burial practices, given the coexistence of cremation and inhumation (a rare thing to capture this transition of cultural practices)

Many thanks for pointing out the strengths of the study. We agree with the reviewer that this is a unique site to study the LBA-IA and the inhumation-cremation transitions. We appreciate your positive feedback.

Weaknesses

1. Limited statistical significance: Many central claims rely heavily on statistically non-significant results. Statements such as:

"Our results show a positive increase in the percentage of steppe-related ancestry between EMBA and FBA, although the increase detected is not significant (Wilcoxon rank-sum test; $p=0.06935$). Using the PC2 coordinates as a proxy for steppe-related ancestry from the same individuals confirms a positive but non-significant trend between the two periods (Wilcoxon rank-sum test; $p = 0.099$)."

Although the authors are careful to mention the insignificance of the statistical tests, they still seem to be drawing conclusions from these tests anyway. This undermines my confidence in the conclusions and I think this manuscript requires a more cautious interpretation and re-writing of many of the conclusions.

Thank you for this comment. We agree that caution is needed when interpreting non-significant statistical trends.

Our intention was not to overstate the results, but rather to highlight a positive trend that becomes statistically significant when comparing the Early Bronze Age (EBA) to the Iron Age (IA) as a whole. This observation was already made by Olalde et al. 2019, but taking Iberia EBA and IA as a whole. Now, including FBA and EIA individuals from the area we note that the temporal shift we describe should not be interpreted as a sudden change at the beginning of the IA, but rather as a gradual process, with Los Castelletts helping to illustrate this continuity. This process only becomes statistically significant at the time of Los Piojos (the new sample from EIA reported for the first time in this manuscript). These revisions are now reflected in the text for greater clarity and precision and a specific paragraph has been added to clarify the genetic trend (lines 354-357) that reads as follows:

"These results, obtained specifically from the northeastern part of the Iberian Peninsula, mirror those obtained by Olalde (2019) when analysing the Peninsula as a whole, while adding finer temporal and geographic resolution."

The authors extensively interpret subtle differences observed in PCA plots. I am not convinced by the trend observed in PCA. The qpAdm models despite acknowledging their limited statistical support.

In light of your comments and the editor's comments, we have removed comments about small location shifts observed only through PCA but not directly quantified by formal f-statistics.

We have removed detailed descriptions of PCA positions from lines 306- 311 that now appears in ~~strikethrough~~ formatting:

~~Los Castelletts II individuals are slightly negatively shifted on PC1 when compared to BA individuals in general. Based on the location on PC2, which correlates well with the proportions of steppe-related ancestry, our new FBA individuals from Los Castelletts II show overall slightly higher PC2 values than EMBA and the one FBA individual published from northeastern Iberia (from Túmulo Mortorum, Cabanes).~~

2. Ambiguity and exploratory models: Multiple qpAdm exploratory models are presented with limited clarity or justification regarding their selection and interpretation. The manuscript acknowledges: "Firstly, we tried to model Los Castelletts II as a mixture... but did not obtain a satisfactory model fit (p-values>0.05)."

Clearer rationale and more explicit discussion on exploratory modeling would enhance the manuscript significantly. I was generally confused by the paragraphs describing all the different models used, I re-read it multiple times but struggled to make sense of why they tested certain models. I think more effort can be made to explain the rationale for the tests used, and maybe some of these results can be moved to a supplementary materials?

In the Results section, we tested the models from the simplest to the most complex, based on prevailing archaeological hypotheses and the territorial boundaries of the Urnfield phenomenon. We have now softened the language and focused on the main results. Some parts have been moved to the Supplementary Material, and others that discussed the findings have been moved to the Discussion section, as suggested by the reviewer. We hope these changes improved the clarity of the reported results, and we are grateful for the suggestion.

3. The discussion contains overly broad statements, like "Our population genomic results document..." These statements would benefit from mentioning which specific analyses they are referring to. Also highlighting limitations and uncertainty more transparently. Especially with all the different models mentioned in the results section, it was confusing what analyses the authors are referring to.

Thank you for this comment. As mentioned earlier, we have now softened the language (e.g., instead of stating 'document', we now say 'suggest'). Additionally, we refer to all our results in relation to our specific site rather than to the Northeastern Iberian region, starting from the first

paragraph of the Discussion. Throughout the Discussion, we continue to highlight that the sample size is limited and that we did not have the opportunity to co-analyze cremated remains using archaeogenomic methods.

We hope that all the adjustments fulfil your standards of transparency in terms of contextual limitations.

4. Sample size limitations: The authors themselves highlight limitations regarding sample size: "A full appraisal of the population dynamics in this region during the FBA is still limited by the data available to date, and several aspects need to be taken with caution." I commend the authors for being up front about this, but it also clearly limits the interpretation of their results. Unfortunately this remains to be a weakness in the manuscript as it stands, and is likely a main reason for the lacking statistical significance.

We agree that the sample size and the archaeological context (e.g., cremations) are still limited for this time period and the Urnfield phenomenon more specifically. However, we are confident that the study is a critical stepping stone: although it includes a limited number of individuals from a very specific context, we are able to rule out invasionist hypotheses formulated in archaeology, at least in relation to this particular site. Overall, we toned down the final Discussion.

Minor Comments

- Figure 1 Issues:

- The legend and map symbols in Figure 1A are inconsistent, making it challenging to interpret accurately. Specifically referring to the line thickness of the stars.

Yes, thank you. We have changed it now and made it consistent.

- Figure 1B's star size is ambiguous. Clarifying whether star diameter or area corresponds to sample size would improve readability. Or perhaps not using stars for the size? The circles are much easier to interpret.

We agree and following your suggestions have changed stars for circles to improve readability.

- Figure 1C shows discrepancies between timeline definitions in the figure and those in the text. Please make these consistent it was a confusing to me.

Thank you. We have changed the figure so that it is consistent with the data presented in the text. Specifically, we have corrected the Late Bronze Age (LBA) coloured period, now covering a chronological time between ~1,400 and ~850 BCE (line 82).

- Clarification of the number of samples included. (very minor point, possible typo)
- Initial statements suggest 25 individuals were analyzed, later adjusted to 24 due to merging duplicated samples: "two samples... showed a pairwise mismatch rate consistent with being sampled from the same individual... thus subsequently merged... treating them as a single individual." But then later (L260) the authors mention 25 individuals used in the final dataset.

Thank you. The final dataset used is 25 individuals, as mentioned in line 262: "In total, we generated genome-wide data from 25 individuals from two different archaeological sites from the northeastern Iberian site of Los Castelletts II (FBA site, n=24) and Los Piojos (EIA site, n=1) (Supplementary Information 1, Supplementary Table S1.1)".

In order to clarify the final number of individuals, in line 276 we changed the word "individuals" to "samples", as follows: "We used BREADR (Rohrlach et al., 2023), KIN (Popli et al., 2023) and READ (Monroy Kuhn et al., 2018) to identify potential duplicate samples". We explain that two samples, LCA005.A (left petrous bone) and LCA009.B (lower left third molar), belong to the same individual, but LCA009.A and LCA009.B (two skeletal elements in principle from the same individual) ended up coming from two different individuals. Please, see the detailed explanation from lines 281 to 286.

- Kinship Analysis:

- The kinship analyses largely confirm expectations (e.g., extended family burial), limiting their novelty somewhat: "This suggests that Tumulus 2 may have been used like a mausoleum by a single extended family..." Am I missing something? I would imagine that a family mausoleum would have family members in it?

Yes, our genetic results confirm the expectations. However, the expectations were based on hypotheses (biological kin-related group) that could not be tested without the quantitative data generated and reported here. While we are aware that kin-related groups do not have to be exclusively biological, genetics help to determine whether such biological relationships exist and provide a critical scaffold for interpreting the remainder inter-individual relations. In this sense, we consider our results to be both novel and highly relevant.

We have removed the term "single" and it now reads as follows:

“This suggests that Tumulus 2 may have been used like a mausoleum by an extended family, where biological ties prevailed over other types of relationships.”

- Results indicating patrilineality lack statistical significance: "...results obtained are not significant due to the presence of some women with relatives at the site (p-value = 0.08)." Again, I realize the manuscript is limited by sample sizes, but this result follows a similar trend of underwhelming statistical results.

Thank you for the comment. We believe the previous sentence did not clearly convey our point: that even with strong indicators of patrilineality, this test yields negative results in other studies (Villalba-Mouco et al. 2023). We have now added a possible explanation (lines 516-519):

“This scenario is similar to the one already observed during the Iberian EMBA, when clear indicators of patrilineal and virilocal practices were found at intra-site level but the statistical test remained non-significant (Villalba-Mouco et al., 2021). A possible explanation for the lack of significance in this test is the inclusion of female individuals below the typical age for exogamy in the cohort of women.”

Nevertheless, we acknowledge that a single tumulus from a single site provides only limited information, which what we intend to address in the respective paragraph (lines 527-531):

“Although there are genetic indications suggesting virilocality in the case of Castellet II, the information from a single funerary tumulus is too limited to determine whether this is indeed the case, or whether we are instead observing a less pronounced trend compared to earlier periods.”

Reviewer #3 (Remarks to the Author):

The paper provides genomic insights into a Late Bronze Age community in north-eastern Iberia, contributing to our understanding of population dynamics, mortuary ritual and mobility patterns of the period. The study will be conducted with scientific rigour and will be an important addition to the bioarchaeological literature. While the paper presents valuable genetic data, some claims require further support: Clarification on cremation analysis (line 46, abstract): The authors should specify on which samples aDNA analyses were performed, given the mixed nature of the context (both cremations and inhumations).

Thanks. We added the word "inhumated" to line 49 to clarify the nature of the samples.

Correct the statement regarding the novelty of the cremation studies (lines 76-77): This statement should be revised to acknowledge the significant advances in cremation studies, including morphological, osteometric and histological investigations. References such as Gigante et al. (2021) should be considered. Clarification of isotopic vs. genomic analyses (line 78): The current presentation of the state of the art is somewhat confusing. A distinction should be made between isotopic and genomic analyses. In addition, the authors should mention that radiogenic Sr isotopes are widely used for studies of individual geographic mobility. Recent publications (e.g. Esposito et al. 2023, Gigante et al. 2025) have successfully applied Sr ratio analysis in contexts where both inhumation and cremation practices coexisted.

Thank you for your recommendations. We have added the bibliographical examples and an additional paragraph with the methodological improvements in the study of cremated individuals (lines 84-89).

Reorganisation of the chronological discussion (line 85): The section introducing chronological abbreviations should be moved earlier in the text to maintain narrative coherence, as the preceding discussion focuses on cremation and bioarchaeological studies.

Thanks. The section introducing chronological abbreviations has been moved upwards as suggested.

Clarification of terminology (line 98): The term "historiographical perspective" does not seem appropriate. "Archaeological perspective" would be a more appropriate alternative.

Thank you. Done!

Distinction between multiple and collective burials (line 173): Throughout the text, the authors use "multiple" and "collective" interchangeably when referring to the presence of several individuals in the same burial space. From a taphonomic perspective, however, these terms have different meanings: "multiple" implies synchronous burials, while "collective" suggests non-contemporaneous burials. The authors should revise the text to clarify the depositional sequence of the individuals studied.

Thank you. We have now changed the word "multiple" for "collective" in line 194. We have also added the definition of "collective" in lines 195-196.

The study has the potential to contribute to ongoing discussions of Final Bronze Age demographic patterns and social organisation in Iberia. However, its impact could be enhanced by a clearer articulation of its unique contributions, particularly with regard to cremation analyses, mobility studies, and the limitations of the dataset. The omission of mobility analysis for cremated individuals should be explicitly mentioned, and potential avenues for future research should be suggested.

We agree that mobility studies need to be explicitly mentioned. Now, they first appear in the Introduction chapter (see the comment above) and in Conclusions which reads as follows:

“Future research could integrate recent Strontium isotope protocols with genetic ancestry data to explore individual mobility in both cremated and non-cremated individuals, thereby offering a more comprehensive understanding of past population dynamics.”

Thank you for all of the suggestions!

Response to referees

We sincerely thank the reviewers and the editor for their insightful and constructive feedback. We have carefully considered all the points raised and revised the manuscript accordingly. Reviewer comments appear in standard black text, followed by our responses in blue italics. All modifications to the manuscript are highlighted in the revised version, with deleted sections indicated using strikethrough formatting.

Reviewers' comments:

Reviewer #1 (Remarks to the Author):

The authors have taken into consideration all suggestions. I think the manuscript is suitable for publication in Communications Biology.

Thank you.

Reviewer #2 (Remarks to the Author):

Thank you for the effort you invested in revising the manuscript. The redesigned figures are clearer, and the slightly re-ordered Result section reads more smoothly. Nonetheless, I still feel that some central conclusions reach a bit beyond what the data securely demonstrate. Below I flag the points that, in my view, most need tightening to bring the narrative fully in line with the evidence.

1. PCA-based inference of Steppe ancestry

The manuscript still states: "Altogether, the PCA suggests an increased level of steppe-related ancestry in Iberian FBA and EIA individuals when compared with EMBA individuals from the same region."

Given the large overlap between EMBA and FBA clouds, I am not yet convinced that the modest shift in PCA space is biologically meaningful. Please either re-phrase to something more cautious (e.g. "is consistent with"), or omit the sentence and let the formal qpAdm / f-statistics carry the claim.

Personally, we think that 'suggest' is a much more cautious expression than 'is consistent with' and this is why the qualitative observation based on PCA placement is followed up with formal testing (quantitative model-based approach) on page 10 using f4-statistics and qpAdm ancestry modelling. However, we agree that the summarising statement in lines 303-305 is not needed that early in the results section and have removed it.

2. Use of non-significant trends

The revised text now reports Wilcoxon p-values of ≈ 0.07 – 0.10 for both Steppe ancestry estimates and PC2 scores, yet still frames these results as evidence of a “gradual increase.”

At conventional thresholds these trends are not statistically significant. Please make this explicit and consider focusing discussion on comparisons that do reach significance.

We agree that Olalde (2019) documents a statistically supported uptick in Steppe-rich ancestry between Bronze- and Iron-Age Iberia and tentatively links it to Urnfield movements. However, those data do not resolve whether the rise began earlier—within the Final Bronze Age—or specifically within the northeastern Urnfield heartland. Your dataset, not Olalde’s, must provide that evidence, ideally with statistics stronger than a $p \approx 0.07$ Wilcoxon.

Thank you for your comments, and we apologize if we did not explain this point clearly enough in the first round of reviews.

Now, we explicitly mention the non-significant trend of increasing Steppe-related ancestry when FBA groups are compared with IA groups populations. In addition, we state that the f_4 -statistic of the form $f_4(\text{Mbuti, test; Russia_Samara_EBA_Yamnaya, Turkey_N})$ is positive, but non-significant. To close the paragraph, now we state:

“Altogether, we cannot rule out the hypothesis that the increase in steppe-related ancestry could have begun already in the FBA. However, additional data from FBA archaeological sites across different geographic regions are needed to document a potentially earlier increase of Steppe-related ancestry and whether it was associated with specific cultural groups.” (see 348-352 lines)

The manuscript is clearly improved, but its two main narrative pillars—(i) a PCA-based Steppe signal within the FBA and (ii) trends hovering just above the significance threshold—still need to be addressed in my opinion.

Thank you again for your revisions. I look forward to reviewing the next iteration.

Thank you, we appreciate your constructive comments. We hope that now we have addressed the points raised satisfactorily.

Reviewer #3 (Remarks to the Author):

The authors have adequately addressed the issues raised in the minor revisions, including the incorporation of the requested bibliographic references. I find that the manuscript is now suitable for publication, and I have no further comments.

Thank you.